# MULTI-AGENT DUAL LEARNING

**Yiren Wang**[†][*], **Yingce Xia**[‡], **Tianyu He**[§], **Fei Tian**[‡], **Tao Qin**[‡], **ChengXiang Zhai**[†], **Tie-Yan Liu**[‡]

[†]University of Illinois at Urbana-Champaign
[‡]Microsoft Research
[§]University of Science and Technology of China
[†]{yiren, czhai}@illinois.edu
[‡]{Yingce.Xia, fetia, taoqin, tie-yan.liu}@microsoft.com
[§]hetianyu@mail.ustc.edu.cn

## ABSTRACT

Dual learning has attracted much attention in machine learning, computer vision and natural language processing communities. The core idea of dual learning is to leverage the duality between the primal task (mapping from domain $\mathcal{X}$ to domain $\mathcal{Y}$) and dual task (mapping from domain $\mathcal{Y}$ to $\mathcal{X}$) to boost the performances of both tasks. Existing dual learning framework forms a system with two agents (one primal model and one dual model) to utilize such duality. In this paper, we extend this framework by introducing multiple primal and dual models, and propose the *multi-agent dual learning* framework. Experiments on neural machine translation and image translation tasks demonstrate the effectiveness of the new framework. In particular, we set a new record on IWSLT 2014 German-to-English translation with a 35.44 BLEU score, achieve a 31.03 BLEU score on WMT 2014 English-to-German translation with over 2.6 BLEU improvement over the strong Transformer baseline, and set a new record of 49.61 BLEU score on the recent WMT 2018 English-to-German translation.

## 1 INTRODUCTION

Motivated by the dual nature of many tasks, e.g., English-to-German vs. German-to-English in machine translation, photo-to-Monet vs. Monet-to-photo in image translation, and speech recognition vs. speech synthesis, dual learning is proposed (He et al., 2016) and applied to many applications including neural machine translation (NMT) (He et al., 2016; Xia et al., 2017a;b), image-to-image translation (Zhu et al., 2017; Yi et al., 2017; Luo et al., 2017), question answering (Duan et al., 2017; Sun et al., 2019) and image captioning (Huang et al., 2018). The basic idea of dual learning is to leverage the duality between the two tasks as the feedback signal to regularize training. The regularization has been implemented with different ways in existing works, such as maximizing the likelihood of data reconstruction (He et al., 2016), constraining joint probabilistic consistency (Xia et al., 2017b), and encouraging the model-level information sharing (Xia et al., 2018). Among them, the principle of likelihood maximization of data reconstruction has been mostly adopted (Zhu et al., 2017; He et al., 2016), in which dual learning is formulated as a two-agent game: One agent targets at learning the mapping $f : \mathcal{X} \mapsto \mathcal{Y}$, while the other learning the mapping of opposite direction $g : \mathcal{Y} \mapsto \mathcal{X}$. Here $\mathcal{X}$ and $\mathcal{Y}$ are two domains. In dual learning, $x \in \mathcal{X}$ is first mapped to $\hat{y} \in \mathcal{Y}$ through $f$ and then reconstructed to $\hat{x} \in \mathcal{X}$ by $g(\hat{y})$. The distortion between $x$ and $\hat{x}$, denoted as $\Delta_x(x, \hat{x})$, is used as the feedback signal to regularize training. Similar reconstruction error can also be constructed in $\mathcal{Y}$ and further used for training.

In such a two-agent game, $g$ and $f$ can be viewed as an evaluator for each other. $g$ is used to evaluate the quality of $\hat{y}$ generated by $f$ and return the feedback signal $\Delta_x(x, g(\hat{y}))$ back to $f$, and vice versa. The quality of such evaluation plays an important role to improve the training of the mapping functions. In the current dual learning framework, only one agent $g$ is used to evaluate and provide feedback signals to the mapping function $f$ in the other direction. In this work, we introduce multiple agents into the learning system to further exploit the potential of dual learning. The agents

---

[*]This work was done when the first author was an intern at Microsoft Research.

in the same direction have similar capability and certain level of diversity to map one domain to the other domain, i.e., $\mathcal{X} \to \mathcal{Y}$ or $\mathcal{Y} \to \mathcal{X}$. Different agents can be obtained by training multiple $f$'s and $g$'s independently with different random seeds for initialization and data access order. Then for the output of each $f$ (or $g$), multiple $g$'s (or $f$'s) will provide feedback signals. Intuitively, more agents can lead to more reliable and robust feedback, just like the majority voting of multiple experts, and it is expected to achieve better final performance. We name this new dual learning framework with multiple agents as *multi-agent dual learning*[1].

Although multiple agents are involved in multi-agent dual learning, we still focus on training two mappings $f_0 : \mathcal{X} \to \mathcal{Y}$ and $g_0 : \mathcal{Y} \to \mathcal{X}$, similar to the traditional two-agent dual learning. The additionally introduced agents play the role of facilitating the training of $f_0$ and $g_0$. The parameters of the additional agents are fixed during training and have no part to play during testing/inference. Therefore, the inference of multi-agent dual learning is of the same complexity as it is of the standard two-agent dual learning. More precisely, apart from the two agents $f_0$ and $g_0$, we further employ $2(N-1)$ additional agents, $N \geq 2$, which are pre-trained before optimizing $f_0$ and $g_0$ and subsequently held constant. Among them, $N-1$ agents are leveraged to model the mapping $f_i : \mathcal{X} \mapsto \mathcal{Y}$ and the others $N-1$ agents model $g_i : \mathcal{Y} \mapsto \mathcal{X}$, $i \in \{1, 2, \cdots, N-1\}$.[2] All these $2(N-1)$ agents are used to facilitate the training of $f_0$ and $g_0$ in a similar manner as the standard dual learning with minor revisions: To help the training of $f_0$, for any $x \in \mathcal{X}$, we first map $x$ to $\hat{y}$ by $f_0$, build the feedback signals by adaptively summing $\Delta_x(x, g_i(\hat{y}))$ for any $i \in \{0, 1, \cdots, N-1\}$, and then regularize training with the feedback signals. The training of $g_0$ is performed similarly.

We conduct experiments on two benchmark tasks of dual learning, NMT and image-to-image translation, to verify our proposed framework. For NMT, we work on the English↔German translation task with different scales of bilingual data. On IWSLT 2014 translation task with $153K$ bilingual data and no monolingual data, we achieve 29.52 and 35.44 BLEU scores for English→German and German→English translations respectively, setting a new record on this task. On WMT 2014 translation task with $4.5M$ bilingual sentence, we achieve 30.05/31.03 and 33.32/35.64 BLEU scores without/with monolingual data, which are the best results of the same settings. On the recent WMT 2018 English→German translation task, we set a new record of 49.61 BLEU score, improving the previous best system by over 1.3 BLEU score. On WMT 2016 unsupervised NMT task, where no bilingual data is available and only $50M$ monolingual data is provided, we achieve 19.26/23.85 BLEU scores, outperforming all the previous pure NMT based systems. We work on two groups of tasks for image-to-image translation, paint↔photo translations (paint including Van Gogh, Monet, Ukiyo-e and Cezanne), and cityscapes label↔photo translation. Experimental results show that we can generate better images than CycleGAN.

The remaining part of this paper is organized as follows. We introduce the basic framework in Section 2. The applications to NMT and image translation are presented in Section 3 and Section 4. We conclude this paper and discuss the future directions in the last section.

## 2 FRAMEWORK

In this section, we mathematically formulate the multi-agent dual learning framework, and compare it with previous learning settings.

Consider two domains of interests, $\mathcal{X}$ and $\mathcal{Y}$. Let $\mathcal{D}_x$ and $\mathcal{D}_y$ denote the collection of training data from $\mathcal{X}$ and $\mathcal{Y}$ respectively, $\mathcal{D}_x \subset \mathcal{X}$, $\mathcal{D}_y \subset \mathcal{Y}$. We aim to learn two agents $f_0 : \mathcal{X} \mapsto \mathcal{Y}$, and $g_0 : \mathcal{Y} \to \mathcal{X}$. Without loss of generality, we refer to learning $f_0$ and $g_0$ as the primal task and the dual task respectively. Let $\Delta_x(x, x')$ be a mapping $\mathcal{X} \times \mathcal{X} \to \mathbb{R}$, representing the dissimilarity/distance/error between $x$ and $x'$ where $x, x' \in \mathcal{X}$, and $\Delta_y(y, y')$ denote the dissimilarity between the two elements

---

[1] Although with the same term, the "multi-agent" or "agent" in this paper has no relationship with multi-agent reinforcement learning

[2] The numbers of agents served for two mapping directions are not necessarily the same. For the tasks where the two domains $\mathcal{X}$ and $\mathcal{Y}$ are fully symmetric, like the language spaces in NMT and image collections in image-to-image translation, learning the two mappings are of similar difficulty. Therefore, we use the same number of agents to assist training in this paper, and leave the study of asymmetric setting to the future work.

$y$ and $y'$ in space $\mathcal{Y}$. The standard dual learning loss (He et al., 2016) is formulated as

$$\frac{1}{|\mathcal{D}_x|} \sum_{x \in \mathcal{D}_x} \Delta_x(x, g_0(f_0(x))) + \frac{1}{|\mathcal{D}_y|} \sum_{y \in \mathcal{D}_y} \Delta_y(y, f_0(g_0(y))), \tag{1}$$

where $|\mathcal{D}_x|$ and $|\mathcal{D}_y|$ denote the number of elements in $\mathcal{D}_x$ and $\mathcal{D}_y$. In a multi-agent dual learning framework, as a prerequisite there are $N - 1$ pretrained primal models $f_i : \mathcal{X} \to \mathcal{Y}$ and $N - 1$ dual models $g_i : \mathcal{Y} \to \mathcal{X}$, $i = \{1, 2, \cdots, N - 1\}$.

Each $f_i$ is pretrained by minimizing the cross entropy loss $-\sum_{(x,y)} \log P(y|x; f_i)$ via stochastic gradient descent where $(x, y)$ is the paired data, or using unsupervised learning techniques to obtain like unsupervised NMT (Lample et al., 2018) or unsupervised image translation (Zhu et al., 2017). Different methods can be used to obtain diverse agents $f_i, i \in \{1, 2, \cdots, N - 1\}$, including using various random seeds to affect weight initialization and input order of the training samples, different model architectures, or training over different subsets. The same holds for $g_i, i \in \{1, 2, \cdots, N-1\}$. In the training process of $f_0$ and $g_0$, all these pretrained $2(N - 1)$ models remain fixed, and are linearly aggregated with $f_0, g_0$ into two mixed models $F_{\boldsymbol{\alpha}}$ and $G_{\boldsymbol{\beta}}$. Specifically, given any $\alpha_i \geq 0$ and $\beta_i \geq 0$ for any $i \in \{0, 1, \cdots, N - 1\}$,

$$F_{\boldsymbol{\alpha}} = \sum_{i=0}^{N-1} \alpha_i f_i, \ G_{\boldsymbol{\beta}} = \sum_{j=0}^{N-1} \beta_j g_j; \ \text{s.t.} \ \sum_{i=0}^{N-1} \alpha_i = 1, \ \sum_{j=0}^{N-1} \beta_j = 1. \tag{2}$$

For multi-agent dual learning, the duality feedback signal is built upon $F_{\boldsymbol{\alpha}}$ and $G_{\boldsymbol{\beta}}$. Following the basic framework of dual learning (He et al., 2016), for any $x \in \mathcal{X}$, all agents first cooperate to generate a $\hat{y} \in \mathcal{Y}$ by $\hat{y} = F_{\boldsymbol{\alpha}}(x)$, and then jointly reconstruct the $\hat{x} \in \mathcal{X}$ through $\hat{x} = G_{\boldsymbol{\beta}}(\hat{y})$. The reconstruction error between $y \in \mathcal{Y}$ and $\hat{y} = F_{\boldsymbol{\alpha}}(G_{\boldsymbol{\beta}}(y))$ is similarly constructed. The out-coming dual learning loss is defined as

$$\ell_{\text{dual}}(\mathcal{D}_x, \mathcal{D}_y; F_{\boldsymbol{\alpha}}, G_{\boldsymbol{\beta}}) = \frac{1}{|\mathcal{D}_x|} \sum_{x \in \mathcal{D}_x} \Delta_x(x, G_{\boldsymbol{\beta}}(F_{\boldsymbol{\alpha}}(x))) + \frac{1}{|\mathcal{D}_y|} \sum_{y \in \mathcal{D}_y} \Delta_y(y, F_{\boldsymbol{\alpha}}(G_{\boldsymbol{\beta}}(y))). \tag{3}$$

In this way, $N$ agent pairs (prima-dual model pairs) are involved to jointly train and improve $f_0$ and $g_0$. When $N = 1$, Our algorithm degenerates to standard dual learning. No labeled information is required for building $\ell_{\text{dual}}$. We can apply it on either labeled data or unlabeled data.

Note that Eqn. (3) is the loss about duality. Other training objectives could also be included. For example, in NMT, if bilingual data is available, the cross entropy loss can be included to guide the training; in image-to-image translation, the GAN loss could also be included to enforce the generated images into the correct categories. In the next two sections, we will discuss how to adapt Eqn. (3) to different applications.

**Discussion**. While there are existing works using multiple agents to boost the model performance, none of them has leveraged the duality. We use the $\mathcal{X} \to \mathcal{Y}$ mapping task to compare the previous work with our proposed framework, following the notations defined in Section 2.

(1) Ensemble learning (Zhou, 2012) is a straightforward way to combine multiple models during inference. To predict the label of $x \in \mathcal{X}$, all agents vote together and the final label would be $\arg\min_{y \in \mathcal{Y}} \sum_{i=0}^{N-1} \alpha_i \ell(f_i(x), y)$, where $\ell$ is the loss function over space $\mathcal{X} \times \mathcal{Y}$. The $\alpha_i$'s can be simply set as $1/N$ or adaptively set according to the quality of each agent. There are several differences between ensemble learning and our work: 1) ensemble learning does not use multiple agents in training as we do; 2) our multi-agent dual learning uses only one model $f_0$ in inference, which is more efficient than ensemble learning that uses multiple agents; and 3) duality is not considered in conventional ensemble learning.

(2) Knowledge distillation with multiple agents (Hinton et al., 2015; Kim & Rush, 2016). Knowledge distillation consists of two steps: first, all $f_i$'s generate soft labels for $x \in \mathcal{X}$, e.g., $\hat{y} = \arg\min_{y \in \mathcal{Y}} \sum_{i=0}^{N-1} \alpha_i \ell(f_i(x), y)$; the generated pairs $(x, \hat{y})$'s are together used to train a new model. Each $(x, \hat{y})$ is regarded as labeled data without evaluating the quality of $\hat{y}$ or considering whether it is good enough for model training. In our proposed framework, we leverage the duality to build a feedback loop so as to evaluate the quality of generated pairs.

# 3 APPLICATION TO NEURAL MACHINE TRANSLATION

In this section, we introduce how to adapt the proposed multi-agent dual learning framework to Neural Machine Translation (NMT), and present evaluation on several public translation datasets.

## 3.1 ADAPTION

Note that for NMT, $\mathcal{X}$ and $\mathcal{Y}$ stands for the collection of all possible sentences of two given languages, while $\mathcal{D}_x$ and $\mathcal{D}_y$ is the dataset. Denote the parameters of mapping $f_0$ and $g_0$ as $\theta_0^f$ and $\theta_0^g$ respectively. Following the common practice in NMT, the $\Delta_x$ and $\Delta_y$ is specified as the negative log-likelihood, that is, for any $x \in \mathcal{D}_x$ or $x \in \mathcal{X}$,

$$\Delta_x(x, G_{\boldsymbol{\beta}}(F_{\boldsymbol{\alpha}}(x))) = -\log \sum_{\hat{y} \in \mathcal{Y}} P(F_{\boldsymbol{\alpha}}(x) = \hat{y}|x; F_{\boldsymbol{\alpha}}, G_{\boldsymbol{\beta}}) P(G_{\boldsymbol{\beta}}(\hat{y}) = x|x, F_{\boldsymbol{\alpha}}(x) = \hat{y}; F_{\boldsymbol{\alpha}}, G_{\boldsymbol{\beta}})$$

$$= -\log \sum_{\hat{y} \in \mathcal{Y}} P(F_{\boldsymbol{\alpha}}(x) = \hat{y}|x; F_{\boldsymbol{\alpha}}) P(G_{\boldsymbol{\beta}}(\hat{y}) = x|\hat{y}; G_{\boldsymbol{\beta}}). \tag{4}$$

For ease of reference, we briefly denote Eqn. (4) as

$$\Delta_x(x, G_{\boldsymbol{\beta}}(F_{\boldsymbol{\alpha}}(x))) = -\log \sum_{\hat{y} \in \mathcal{Y}} P(\hat{y}|x; F_{\boldsymbol{\alpha}}) P(x|\hat{y}; G_{\boldsymbol{\beta}}). \tag{5}$$

Similarly, for any $y \in \mathcal{Y}$, we have

$$\Delta_y(y, F_{\boldsymbol{\alpha}}(G_{\boldsymbol{\beta}}(y))) = -\log \sum_{\hat{x} \in \mathcal{X}} P(\hat{x}|y; G_{\boldsymbol{\beta}}) P(y|\hat{x}; F_{\boldsymbol{\alpha}}). \tag{6}$$

Following Hassan et al. (2018), we minimize the upper bounds of $\Delta_x$ and $\Delta_y$ for ease of optimization, which are respectively denoted as $\bar{\Delta}_x$ and $\bar{\Delta}_y$:

$$\bar{\Delta}_x(x, G_{\boldsymbol{\beta}}(F_{\boldsymbol{\alpha}}(x))) = -\sum_{\hat{y} \in \mathcal{Y}} P(\hat{y}|x; F_{\boldsymbol{\alpha}}) \log P(x|\hat{y}; G_{\boldsymbol{\beta}}) \geq \Delta_x(x, G_{\boldsymbol{\beta}}(F_{\boldsymbol{\alpha}}(x)));$$

$$\bar{\Delta}_y(y, F_{\boldsymbol{\alpha}}(G_{\boldsymbol{\beta}}(y))) = -\sum_{\hat{x} \in \mathcal{X}} P(\hat{x}|y; G_{\boldsymbol{\beta}}) \log P(y|\hat{x}; F_{\boldsymbol{\alpha}}) \geq \Delta_y(y, F_{\boldsymbol{\alpha}}(G_{\boldsymbol{\beta}}(y))). \tag{7}$$

where two $\geq$'s hold due to Jensen's inequality. Then we turn to minimize

$$\tilde{\ell}_{\text{dual}}(\mathcal{D}_x, \mathcal{D}_y; F_{\boldsymbol{\alpha}}, G_{\boldsymbol{\beta}}) = \frac{1}{|\mathcal{D}_x|} \sum_{x \in \mathcal{D}_x} \bar{\Delta}_x(x, G_{\boldsymbol{\beta}}(F_{\boldsymbol{\alpha}}(x))) + \frac{1}{|\mathcal{D}_y|} \sum_{y \in \mathcal{D}_y} \bar{\Delta}_y(y, F_{\boldsymbol{\alpha}}(G_{\boldsymbol{\beta}}(y))). \tag{8}$$

Directly calculating the gradients for $\bar{\Delta}_x$ w.r.t $\theta_0^f$ and $\theta_0^g$ is difficult in that: i) $\mathcal{Y}$ is exponentially large since it represents all possible sentences of a language, which makes the computation intractable . ii) All $f_i$'s and $g_i$'s ($i \geq 1$) need to be loaded into the memory, although they will not be updated, which makes the computation inefficient. Similar issues exist for computing the gradients of $\bar{\Delta}_y$ w.r.t $\theta_0^f$ and $\theta_0^g$.

We use importance sampling to address the above two difficulties. Define $\boldsymbol{\gamma} = (0, \frac{1}{N-1}, \cdots, \frac{1}{N-1})$, thus $F_{\boldsymbol{\gamma}}$ and $G_{\boldsymbol{\gamma}}$ represent the combined model from all the pre-trained agents without the target models $f_0$ and $g_0$ (see Eqn. (2)). Define

$$\delta(x, \hat{y}) = \left(\frac{P(\hat{y}|x; F_{\boldsymbol{\alpha}})}{P(\hat{y}|x; F_{\boldsymbol{\gamma}})}\right) \log P(x|\hat{y}; G_{\boldsymbol{\beta}}), \ \phi(y, \hat{x}) = \left(\frac{P(\hat{x}|y; G_{\boldsymbol{\beta}})}{P(\hat{x}|y; G_{\boldsymbol{\gamma}})}\right) \log P(y|\hat{x}; F_{\boldsymbol{\alpha}}). \tag{9}$$

Then the gradients for $\bar{\Delta}_x$ w.r.t $\theta_0^f$ is: $\frac{\partial \bar{\Delta}_x}{\partial \theta_0^f} = -\sum_{\hat{y} \in \mathcal{Y}} P(\hat{y}|x; F_{\boldsymbol{\gamma}}) \frac{\partial \delta(x, \hat{y})}{\partial \theta_0^f}$, and can be estimated by:

(i) Given an $x \in \mathcal{D}_x$ and $y \in \mathcal{D}_y$, sample a $\hat{y}$ according to the distribution $P(\cdot|x; F_{\boldsymbol{\gamma}})$; sample an $\hat{x}$ according to the distribution $P(\cdot|y; G_{\boldsymbol{\gamma}})$;

(ii) Estimate the gradients of $\bar{\Delta}_x$ w.r.t $\theta_0^f$ as follows (and similarly for $\theta_0^g$ without loss of generality):

$$\frac{\partial \bar{\Delta}_x}{\partial \theta_0^f} = \mathbb{E}_{\hat{y} \sim P(\cdot|x; F_{\boldsymbol{\gamma}})} \left[ -\frac{\partial \delta(x, \hat{y})}{\partial \theta_0^f} \right] \approx -\frac{\partial \delta(x, \hat{y})}{\partial \theta_0^f}. \tag{10}$$

Our method is summarized in Algorithm 1. We do the offline sampling by sampling $\hat{x}$'s and $\hat{y}$'s in advance (step 3); and computing the probabilities related to $f_i$'s and $g_i$'s $i \geq 1$ in advance (step 4). When bilingual data is available, denoted as $\mathcal{B} = \{(x_k, y_k)\}_{k=1}^{M}$ where $M$ is the number of training data, we can also apply the negative log-likelihood loss on bilingual data.

---

**Algorithm 1:** Algorithm for multi-agent dual learning.

---

1   **Input:** Data $\mathcal{D}_x$ and $\mathcal{D}_y$; learning rate $\eta$; $f_i$ and $g_i$ $i \in \{0, 1, \cdots, N-1\}$; mini-batch size $K$; bilingual data $\mathcal{B}$ if possible;

2   Define $\gamma = (0, \frac{1}{N-1}, \cdots, \frac{1}{N-1})$ and $\alpha = \beta = (\frac{1}{N}, \frac{1}{N}, \cdots, \frac{1}{N})$;

3   Sample the datasets $\mathcal{D}_x$ and $\mathcal{D}_y$; obtain $\hat{\mathcal{D}}_y = \{\hat{y} \sim F_\gamma(x) | x \in \mathcal{D}_x\}$, $\hat{\mathcal{D}}_x = \{\hat{x} \sim G_\gamma(y) | y \in \mathcal{D}_y\}$;

4   Compute the probabilities $P(\hat{y}|x; F_\gamma)$ and $P(\hat{x}|y; G_\gamma)$;

5   **while** *not converged* **do**

6      Randomly sample two batches of $\mathcal{B}_{x\hat{y}} \subset \mathcal{D}_x \times \hat{\mathcal{D}}_y$ and $\mathcal{B}_{\hat{x}y} \subset \hat{\mathcal{D}}_x \times \mathcal{D}_y$, each of size $K$;

7      Following Eqn. (9) and Eqn. (10), calculate $\mathrm{Grad}_{f_0}$ and $\mathrm{Grad}_{g_0}$ as follows:

       $\mathrm{Grad}_{f_0} \leftarrow -\frac{1}{|\mathcal{B}_{x\hat{y}}|} \sum_{(x,\hat{y}) \in \mathcal{B}_{x\hat{y}}} \frac{\partial \delta(x,\hat{y})}{\partial \theta_0^f} - \frac{1}{|\mathcal{B}_{\hat{x}y}|} \sum_{(\hat{x},y) \in \mathcal{B}_{\hat{x}y}} \frac{\partial \phi(y,\hat{x})}{\partial \theta_0^f}$ and

       $\mathrm{Grad}_{g_0} \leftarrow -\frac{1}{|\mathcal{B}_{x\hat{y}}|} \sum_{(x,\hat{y}) \in \mathcal{B}_{x\hat{y}}} \frac{\partial \delta(x,\hat{y})}{\partial \theta_0^g} - \frac{1}{|\mathcal{B}_{\hat{x}y}|} \sum_{(\hat{x},y) \in \mathcal{B}_{\hat{x}y}} \frac{\partial \phi(y,\hat{x})}{\partial \theta_0^g}$;

8      If bilingual data is available, sample a batch $\mathcal{B}_{xy} \subset \mathcal{B}$ of size $K$, calculate

       $\mathrm{Grad}_{f_0} \leftarrow \mathrm{Grad}_{f_0} - \frac{1}{K} \nabla_{\theta_0^f} \sum_{(x,y) \in \mathcal{B}_{xy}} \log P(y|x; f_0)$ and

       $\mathrm{Grad}_{g_0} \leftarrow \mathrm{Grad}_{g_0} - \frac{1}{K} \nabla_{\theta_0^g} \sum_{(x,y) \in \mathcal{B}_{xy}} \log P(x|y; g_0)$;

9      Update the parameters: $\theta_0^f \leftarrow \theta_0^f - \eta \, \mathrm{Grad}_{f_0}$, $\theta_0^g \leftarrow \theta_0^g - \eta \, \mathrm{Grad}_{g_0}$;

10   **end**

---

## 3.2   EXPERIMENT SETTINGS

**Dataset** We use multiple benchmark NMT datasets to evaluate the effectiveness of the proposed framework, including IWSLT 2014 English↔German translation, WMT 2014 English↔German translation and WMT 2016 unsupervised English↔German translation. English and German is denoted as "En" and "De" respectively for ease of reference. For IWSLT 2014 En↔De translation, following Edunov et al. (2018b), we lowercase all the sentences, and split them into training/validation/test set with $153k/7k/7k$ sentences respectively. For WMT 2014 En↔De translation, we choose WMT 2014 training set and filter out $4.5M$ sentences pairs following Gehring et al. (2017) and Vaswani et al. (2017). We concatenate newstest 2012 and newstest 2013 as the validation set and use newstest 2014 as the test set. We also select $8M$ English and $8M$ German monolingual sentences from newscrawl 2013 as monolingual dataset. For unsupervised En↔De, following Lample et al. (2018), we choose 50M monolingual English and German sentences and use newstest 2016 as the test set for a fair comparison with the previous work. We preprocess the words into word-pieces in the same way as Wu et al. (2016).

**Model Architecture** We use Transformer (Vaswani et al., 2017) as the basic model structure. For IWSLT En↔De, we use the *transformer_small* configuration with 4 and 8 blocks to verify the generality of our model with different structures (a 4-block setting refer to transformer with 4 encoder blocks and 4 decoder blocks), where the word embedding dimension, hidden state dimension and non-linear layer dimension is set to be 256, 256 and 1024 respectively. For the WMT task, following Vaswani et al. (2017), we use the *transformer_big* setting with 6 blocks, where the word embedding dimension, hidden state dimension and non-linear layer dimension is 1024, 1024 and 4096 respectively. For the unsupervised NMT task, we choose *transformer_base* setting following Lample et al. (2018), where the above three dimensions are 512, 512 and 2048. The dropout rates for the three settings are 0.2, 0.1 and 0.1 respectively.

**Optimization and Evaluation** We choose Adam (Kingma & Ba, 2015) to optimize the network. The $\eta$ in Algorithm 1 is set as $2 \times 10^{-4}$. The learning rate decay rule and the two $\beta$'s of Adam are the same as Vaswani et al. (2017). For the three tasks, we use one, eight and four M40 GPUs to train those networks for three, five and six days respectively. Beam search is applied to generate translations for all the models, where the beam sizes for the three tasks are 6, 4 and 4 respectively.

The evaluation metric is BLEU score (Papineni et al., 2002), which is the geometric mean of $n$-gram precisions, $n \in \{1, 2, 3, 4\}$.

**Baselines** We implement three types of important baselines[3], including back translation (Sennrich et al., 2015; Edunov et al., 2018a), knowledge distillation (Kim & Rush, 2016) and the two-agent dual learning (He et al., 2016). Take the $f : \mathcal{X} \mapsto \mathcal{Y}$ translation as an example: 1) To use back translation, a reversed translation model $g : \mathcal{Y} \mapsto \mathcal{X}$ is first trained on the original bilingual dataset, then the dataset $\{(g(y), y) | y \in \mathcal{D}_y\}$ is constructed, which is then concatenated with the original dataset and train another model $f$. 2) To use knowledge distillation, we pre-train a teacher model $f_T : \mathcal{X} \rightarrow \mathcal{Y}$, and use it to generate the corresponding dataset $\{(x, f_T(x)) | x \in \mathcal{D}_x\}$, which is then mixed with the ground truth data and used to train a new model[4]. For these two baselines, we generate sentences using both a single model and multiple models. 3) Two-agent dual learning, which uses $f$ together with $g$ to build the feedback signal to regularize the training (see Eqn. (1)), which we denote as "Dual-1". Our multi-agent models are denoted as "Dual-$i$" ($i > 1$)

## 3.3    RESULTS OF NMT ON IWSLT DATASET

We regard the English and German sentences in the IWSLT14 bilingual corpus as $\mathcal{D}_x$ and $\mathcal{D}_y$ in Algorithm 1. We use 4-block networks (i.e. $F_\gamma$ in Algorithm 1) to generate translations, which are then used to train both 4-block (4B) and 8-block (8B) networks. We compare dual learning (Dual-$i$) with the standard baseline (Standard), knowledge distillation (KD-$i$) and back translation (BT-$i$) with different number of primal-dual model pairs $i \in \{1, 5\}$. Specifially, Dual-1 is the two-agent dual learning baseline, and Dual-5 is our multi-agent model with 4 additional pairs of agents.

The results of IWSLT En↔De are presented in Table 1. We can observe that: 1) Dual learning has brought significant improvement over all the baselines (Standard, KD and BT) in both single-agent and multi-agent settings, demonstrating the effectiveness of dual learning; 2) Involving multiple agents into the learning system leads to better performances, which shows the importance of additional feedback signals from other agents. In particular, Dual-5 outperforms Dual-1 by around $0.5$ BLEU in 4B setting and around $0.8$ BLEU in 8B setting, which proves that dual learning can benefit from cooperating with more agents and demonstrates the effectiveness of our proposed algorithm. Particularly, dual learning with $N = 5$ agent pairs under 8B setting sets a new record of $35.44$ BLEU in De→En and $29.52$ BLEU in En→De translations.

Table 1: BLEU scores on IWSLT De↔En translation. "KD", "BT" and "Dual" stands for knowledge distillation, back translation and dual learning respectively.

|  | Standard | KD-1 | KD-5 | BT-1 | BT-5 | Dual-1 | Dual-5 |
|---|---|---|---|---|---|---|---|
| De→En (4B) | 33.42 | 33.89 | 34.20 | 33.71 | 33.61 | 34.25 | **34.70** |
| En→De (4B) | 27.89 | 28.45 | 28.65 | 28.35 | 28.22 | 28.63 | **28.99** |
| De→En (8B) | 34.01 | 34.36 | 34.85 | 33.87 | 33.77 | 34.57 | **35.44** |
| En→De (8B) | 27.95 | 28.74 | 29.18 | 28.28 | 28.25 | 29.07 | **29.52** |

**Study on different number of agents**. We study the performances of our proposed algorithm with respect to different numbers of agent pairs, i.e., $N$ and explore the optimal value of $N$. We enumerate within $N = \{1, \cdots, 5\}$ and did not try larger $N$ values due to the limitation of computational resources. As we can see from Figure 1, more agents can bring better performance for all settings. While it appears that the performance becomes better with more pair of agents involved into the learning system, the gain becomes more and more marginal (e.g., the maximum difference for $N = 4$ and $N = 5$ across all settings is around $0.1$ BLEU). Furthermore, larger $N$ leads to higher computational costs such as GPU resources. Thus, it is the most practical to leverage $N = 3$ agent pairs, where we can benefit from the substantial gain over baselines without too much computational costs. We employ $N = 3$ for the rest of our experiments.

---

[3]All the implementations are based on the official tensor2tensor release: `https://github.com/tensorflow/tensor2tensor`.

[4]We have tried both mixing with ground truth and not and found that mixing is better. On IWSLT 2014 De→En, with/without mixing can achieve 33.89/33.14 BLEU scores.

**Study on diversity of agents**. We further study the influence of diversity among the agents. We show that diversity is important but not the focus of our algorithm. Detailed results and discussion is presented in Appendix A due to the space limitation.

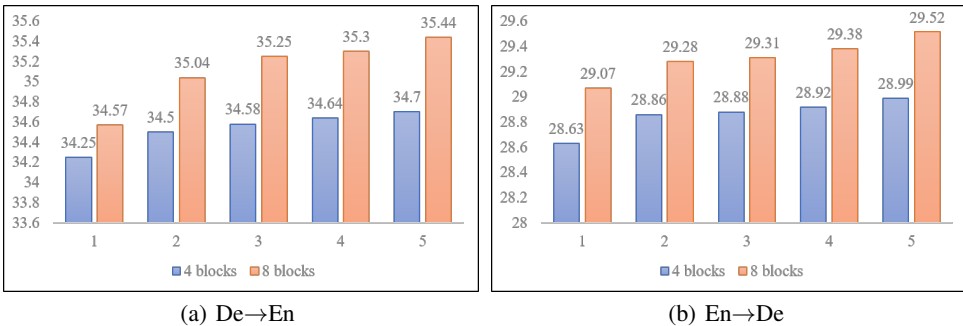

(a) De→En        (b) En→De

Figure 1: BLEU scores on IWSLT De↔En w.r.t the number of agents pairs.

**Study on translation tasks for other language pairs**. We carry out another three groups of IWSLT14 experiments, English↔Spanish (Es), English↔Russian (Ru) and English↔Hebrew (He) translations to verify the generality of our framework on different languages. As is illustrated in Table 2, we achieve consistent improvements across different translation tasks (for all languages and all translation directions), further demonstrating the effectiveness and robustness of our algorithm.

Table 2: BLEU scores on the translations between {Es, Ru, He} and En.

|  | Es→En | En→Es | Ru→En | En→Ru | He→En | En→He |
|---|---|---|---|---|---|---|
| Standard | 40.50 | 37.95 | 18.94 | 16.06 | 33.25 | 22.20 |
| Dual-1 | 42.14 | 38.09 | 21.14 | 16.31 | 35.42 | 22.75 |
| Dual-3 | 42.41 | 38.62 | 21.56 | 16.71 | 36.08 | 23.80 |

## 3.4 RESULTS OF NMT ON WMT DATASET

We work with two settings on WMT En↔De task, where (1) only bilingual data is available; and (2) additional monolingual data is provided. The results are summarized in Table 3.

Table 3: BLEU scores on WMT14 En↔De translation. "Bitext" and "Mono" respectively represents using bilingual data only and using the mix of bilingual&monolingual data.

|  | Standard | KD-1 | KD-3 | BT-1 | BT-3 | Dual-1 | Dual-3 |
|---|---|---|---|---|---|---|---|
| En→De (Bitext) | 28.4 | 29.17 | 29.28 | 28.26 | 27.79 | 29.44 | **30.05** |
| De→En (Bitext) | 32.15 | 32.43 | 31.38 | 32.41 | 32.68 | 32.99 | **33.32** |
| En→De (Mono) | 28.4 | 29.20 | 29.36 | 29.42 | 29.68 | 29.93 | **31.03** |
| De→En (Mono) | 32.15 | 32.28 | 32.43 | 34.58 | 34.77 | 34.98 | **35.64** |

From this table, we can see that the proposed algorithm achieves the best performances for all settings, and we observe that: 1) With WMT 2014 bilingual data only, the proposed multi-agent dual learning (Dual-3) achieves the state-of-the-art results to our knowledge, with 30.05 BLEU for En→De and 33.32 BLEU for De→En[5]. 2) Compared with the traditional two-agent dual learning in He et al. (2016), we can achieve 0.61 BLEU score improvement on En→De and 0.33 on De→En with bilingual data only. For the setting of using monolingual data, the improvements are 1.10 and 0.66 respectively. 3) Although KD and BT can bring improvements under different settings, the

---

[5]Although Edunov et al. (2018a) provides a higher BLEU score, they use WMT'18 dataset, which is different from ours and currently not widely used for WMT14 En↔De tasks in NMT literature.

results are not as consistent as dual learning. For example, the performance of BT is not as good as KD under bilingual setting, but is better than KD under monolingual setting. We conjecture the reason is that with feedback signals to a generated sentence in the dual learning framework, the dataset is not only enlarged through sampling, but also guaranteed with good quality of generation.

**Study on generality of the algorithm**. We apply our method to the state-of-the-art systems on WMT 2016, 2017 and 2018 English-to-German translation challenge to verify the generality of our method. We work with the degenerated case where $f_0$ is warm started with a state-of-the-art model provided by FAIR[6] and $g_0$ is fixed. We adopt the commercial system Google Translator (crawled on Oct 16, 2018), the champion system of WMT18 En→De: MS-Marian (Junczys-Dowmunt, 2018), and FAIR's single and ensemble models (Edunov et al., 2018a) as our baselines. The models are evaluated by sacreBLEU[7]. More details are left in Appendix B.

As can be seen from Table 4, our method is capable of further improving the strong model by a large margin. Both single and ensemble models with our algorithm outperform the previous best systems and set state-of-the-art results on these tasks, demonstrating the effectiveness of our method.

Table 4: BLEU scores on WMT {16, 17, 18} En→De. The single models are independently trained for six times, with mean and standard derivation values reported; the ensemble models are the ensemble results of all different runs.

|  | 2016 | 2017 | 2018 |
|---|---|---|---|
| Google Translator | 38.03 | 31.41 | 47.67 |
| FAIR (Single) | $37.04 \pm 0.16$ | $31.86 \pm 0.21$ | $44.63 \pm 0.12$ |
| **Ours (Single)** | $\mathbf{40.68 \pm 0.11}$ | $\mathbf{33.47 \pm 0.16}$ | $\mathbf{48.89 \pm 0.13}$ |
| MS-Marian (Ensemble) | 39.6 | 31.9 | 48.3 |
| FAIR (Ensemble) | 37.99 | 32.80 | 46.05 |
| **Ours (Ensemble)** | **41.23** | **34.01** | **49.61** |

## 3.5 Results of NMT with Monolingual Data Only

Unsupervised NMT is studied recently to learn two translation models without bilingual data (details in Appendix C). We pre-train two unsupervised NMT models with different initialization, use them to translate the $50M$ monolingual sentences, and apply KD, BT and dual learning algorithms.

Table 5: BLEU scores on WMT 2016 unsupvised NMT En↔De translation.

|  | Standard | KD-1 | KD-3 | BT-1 | BT-3 | Dual-1 | Dual-3 |
|---|---|---|---|---|---|---|---|
| En→De | 17.52 | 17.33 | 18.27 | 17.50 | 18.15 | 18.01 | **19.26** |
| De→En | 22.12 | 22.10 | 22.71 | 22.51 | 22.91 | 22.17 | **23.85** |

As is shown in Table 5, with one pair of agents, KD-1 and BT-1 obtains almost the same results as the standard baseline, while Dual-1 achieves slightly better results on En→De translation. When we increase the number of agents, there are substantial improvements for KD-3, BT-3 and Dual-3. Especially, multi-agent dual learning (Dual-3) achieves 19.26 and 23.85 BLEU scores, setting new records for unsupervised NMT with pure NMT models.

## 4 Application to Image Translation

In this section, We apply the multi-agent dual learning framework to image translation tasks. Different from Ciregan et al. which also uses multiple agents (several columns of deep neural networks), we leverage the duality of translation tasks and only use one model during inference. We follow the setting of CycleGAN (Zhu et al., 2017), the most popular implementation of image translation that combines the ideas of GAN and dual learning.

---

[6]https://s3.amazonaws.com/fairseq-py/models/wmt18.en-de.ensemble.tar.bz2

[7]sig=BLEU+case.mixed+lang.en-de+numrefs.1+smooth.exp+test.wmt{16,17,18}+tok.13a+version.1.2.11

## 4.1 ADAPTION

Following the common practice in image translation (Zhu et al., 2017), for each term in Eqn. (3), the $\Delta_x$ and $\Delta_y$ are specified as the $L1$ norm of pixel-level difference between two images. We set $\alpha = \beta = (\frac{1}{N}, \frac{1}{N}, \cdots, \frac{1}{N})$. Since there is few literature focusing on generating consensus images using multiple models, we switch Eqn. (3) to a simpler form:

$$\ell_{\text{dual}}(\mathcal{D}_x, \mathcal{D}_y; F_{\boldsymbol{\alpha}}, G_{\boldsymbol{\beta}}) = \frac{1}{|\mathcal{D}_x|} \sum_{x \in \mathcal{D}_x} \|x - G_{\boldsymbol{\beta}}(f_0(x))\|_1 + \frac{1}{|\mathcal{D}_y|} \sum_{y \in \mathcal{D}_y} \|y - F_{\boldsymbol{\alpha}}(g_0(y))\|_1, \quad (11)$$

where $\mathcal{D}_x$ and $\mathcal{D}_y$ are the datasets of domain $\mathcal{X}$ and domain $\mathcal{Y}$. The remaining parts of our model, including the model components, the objective functions, are exactly the same as the standard Cycle-GAN. There are two discriminators $d_{\mathcal{X}} : \mathcal{X} \mapsto [0, 1]$ and $d_{\mathcal{Y}} : \mathcal{Y} \mapsto [0, 1]$ used to differentiate the generated images from natural images. The outputs of $d_{\mathcal{X}}$ and $d_{\mathcal{Y}}$ represent the probability that the input image is a natural one. The GAN loss (Goodfellow et al., 2014) is defined as

$$\ell_{\text{GAN}} = \frac{1}{|\mathcal{D}_x|} \sum_{x \in \mathcal{X}} [\log d_{\mathcal{X}}(x) + \log\left(1 - d_{\mathcal{Y}}(f_0(x))\right)] + \frac{1}{|\mathcal{D}_y|} \sum_{y \in \mathcal{Y}} [\log d_{\mathcal{Y}}(y) + \log\left(1 - d_{\mathcal{X}}(g_0(y))\right)].$$

During the optimization process, the two discriminators will try to maximize $\ell_{\text{GAN}}$, while the $f_0$ and $g_0$ will minimize $\ell_{\text{dual}}(\mathcal{D}_x, \mathcal{D}_y; F_{\boldsymbol{\alpha}}, G_{\boldsymbol{\beta}}) + \lambda \ell_{\text{GAN}}$ in cooperation with other provided agents, where $\lambda$ is a hyper-parameter to be tuned. In the experiments, we follow Zhu et al. (2017) to set $\lambda$ as 10.

## 4.2 SETTINGS

We use the same datasets, model configuration and optimization algorithm as Zhu et al. (2017). We implement our method based on CycleGAN[8]. We choose two tasks, the photo↔painting translations where the art styles include Monet, Van Gogh, Ukiyo-e, Cezanne, and cityscapes label↔photo translation. All the images are cropped to $256 \times 256$. The configuration details can be found at Appendix D.1, which follows the same settings as Zhu et al. (2017).

## 4.3 RESULTS OF PAINTING-TO-PHOTO TRANSLATION

The task of painting→photo translation is to translate an input painting with specific art style to the original photo captured by a camera. We present case studies on painting→photo translations in Figure 2. Indeed, there are several cases that our algorithm is clearly better than CycleGAN: For the images in the top-left corner, CycleGAN fails to recover the tree while our algorithm succeeds, with many details correctly included like the branches and leaves. In the bottom right corner, the texture of the ground is kept by our results, which is missing from the images generated by CycleGAN.

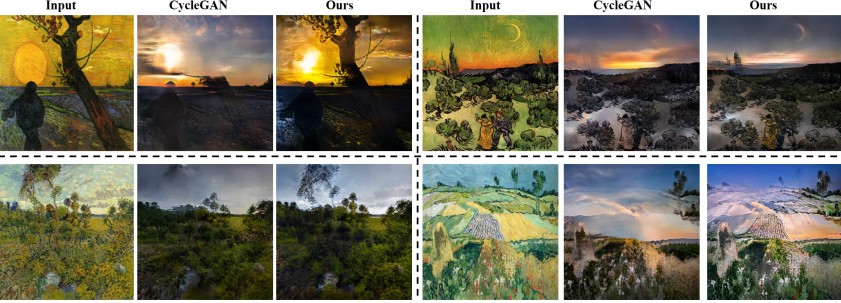

Figure 2: Four groups of experiments on painting→photo translations.

We further present quantitative evaluation with Fréchet Inception Distance (briefly, FID), which is introduced in Heusel et al. (2017) and verified to be a reasonable metric by Lucic et al. (2018), to evaluate the quality of painting→photo translations. FID captures the distance between generated images and the real ones, and a smaller FID score indicates the better model quality. As is shown in Table 6, our method yields consistent improvements across all datasets. More results in photo→painting are left in Appendix D.2 due to space limitation.

---

[8]https://github.com/junyanz/pytorch-CycleGAN-and-pix2pix

Table 6: FID scores on painting→photos translation.

|  | Cezanne | Monet | Ukiyo-e | Vangogh |
|---|---|---|---|---|
| CycleGAN | 189.40 | 134.25 | 197.30 | 91.19 |
| Ours | 173.92 | 129.86 | 184.69 | 89.32 |

## 4.4 RESULTS OF CITYSCAPES LABEL-TO-PHOTO TRANSLATION

We present case studies and quantitative evaluation of label→photo translation on cityscapes images in Figure 3 and Table 7. As can be seen from Figure 3, both CycleGAN and our method can transfer the input label to a matching natural scene, while results of ours are clearer with less noise. For example, in the top left corner, our method can generate flatten roads, clearer cars and buildings and the baseline model fails. In Table 7, we present quantitative evaluation with "FCN-scores" following Isola et al. (2017). Our method achieves better results on the three evaluation metrics. More details about FCN-score and results on photo→label translation are left in Appendix D.3.

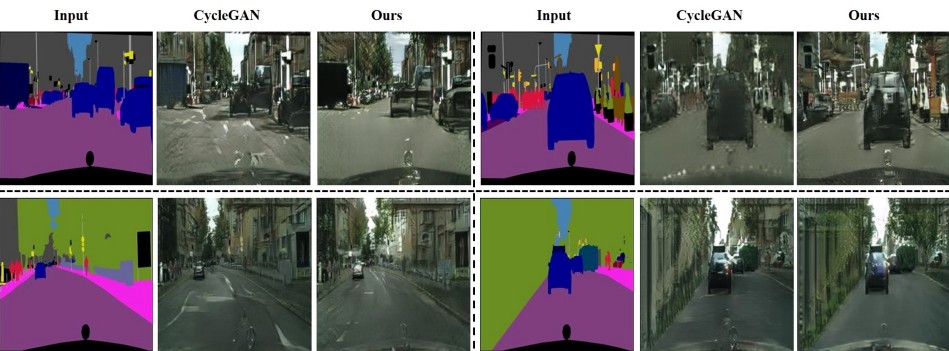

Figure 3: Cityscapes labels → photos translation.

Table 7: FCN-scores on cityscapes labels→photos translation.

|  | Per-pixel Acc. | Per-class Acc. | Class IOU |
|---|---|---|---|
| CycleGAN | 0.52 | 0.17 | 0.11 |
| Ours | 0.57 | 0.19 | 0.14 |

## 5 CONCLUSION

In this paper, we proposed a new framework, multi-agent dual learning, in which multiple primal models and dual models are involved in the learning process. We empirically verified the effectiveness of the proposed framework on multiple machine translation tasks and image translation tasks.

Extending the multi-agent dual learning framework to more applications would be an interesting direction for future research. For example, we can apply our framework to improve the recent non-autoregressive machine translation, which is known to have high inference efficiency yet poor translation quality (Gu et al., 2018; Wang et al., 2019; Guo et al., 2019). It is also worth studying how to apply the framework to other dual learning paradigms, including dual supervised learning (Xia et al., 2017b) and dual transfer learning (Wang et al., 2018). Furthermore, it is challenging yet important to improve the training efficiency while maintaining the substantial improvements.

### ACKNOWLEDGEMENT

This work is partially supported by the National Science Foundation under Grant No. 1801652.

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

## A STUDY ON DIVERSITY OF AGENTS

We further study how the diversity among agents influences the performances of our algorithm on the IWSLT De↔En dataset. We compare three group of agents: **(1) Agents with independent runs ("*Independent Run*")**: We use the same agents as Section 3.3, which have the same model architecture and hyperparameters, and are trained through independent runs with different random seeds. **(2) Agents with different architectures ("*Different Arch*")**: The agents have different hidden dimensions $(256, 512)$ with different numbers of heads $(2, 4)$. They are also trained through independent runs with different random seeds. Intuitively, this group of agents with different architectures could be more diverse than the first group. **(3) Agents from a same run ("*Homogeneous*")**: In comparison to the first two groups where different methods (i.e., different random seeds, different model architecture) are used to encourage diversity among agents, in this group, we use model checkpoints at different (but very close) iterations from a same run. At the late stage of training, the learning rate becomes smaller. Thus the diversity among agents would naturally be very poor.

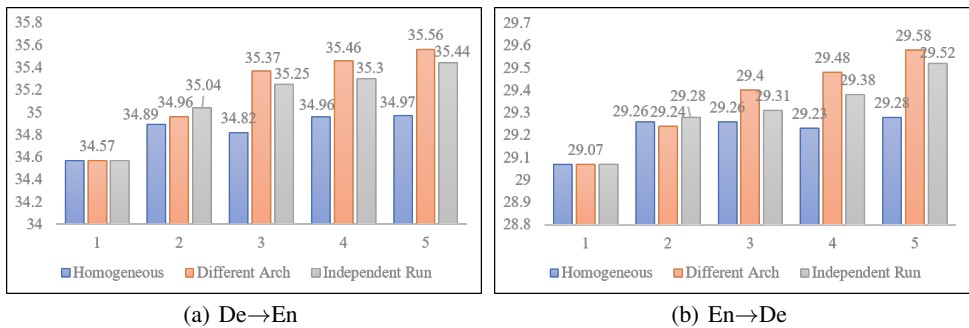

(a) De→En          (b) En→De

Figure 4: BLEU scores on IWSLT De↔En with different pools of agents.

We train 8-block (8B) networks as Section 3.3 with $N = \{1, ..., 5\}$ pairs of agents from the above three groups respectively. The same training and inference procedures as Section 3.3 are applied here. As is shown from results in Figure 4, we have the following observations and conclusions:

1. Diversity among agents plays an important role in our method. The *Homogeneous* agents with poor diversity has the worst performances among the three groups. That is, introducing more homogeneous agents brings very marginal gain.

2. A more diverse group of agents can potentially bring further improvements, yet diversity is not the key focus of our algorithm. *Independent Run* achieves comparable performances with *Different Arch*, suggesting that our framework can achieve substantial improvements with a reasonable level of diversity among agents.

## B DETAILS ON WMT18 EN→DE CHALLENGES

For the study on generality of the algorithm on WMT18 En→De challenges, the training data, $F_\gamma(\cdot)$ and $G_\gamma(\cdot)$ are the same as those used in Section 3.4. We work with a degenerated case where $g_0$ (i.e., the De→En model) is pre-trained before optimizing $f_0$ and subsequently hold constant. The $f_0$ is warm started by the models from `https://dl.fbaipublicfiles.com/fairseq/models/wmt18.en-de.ensemble.tar.bz2`, which are well trained with tremendous resources. We degenerate our algorithm to optimize the single direction due to the limitation that only En→De model checkpoints are released.

We train the model on 8 M40 GPUs for 2 days with PyTorch implementation of our algorithm based on FairSeq Toolkit[9] . The batch size is 4096 tokens. We use Adam optimizer and the learning rate is automatically calculated by the system based on the reloaded checkpoint. The dropout ratio is 0.3. We use beam search with beam size 5 to generate candidates.

---

[9]`https://github.com/pytorch/fairseq`

## C   INTRODUCTION TO UNSUPERVISED NMT

We briefly introduce the recent state-of-the-art unsupervised NMT algorithm in (Lample et al., 2018). Similar to the standard NMT model with bilingual training data, an unsupervised NMT model is also based on the encoder-decoder architecture. The source sentences and target sentences are mapped into a same vocabulary using BPE techniques. The shared embedding is pretrained with fastText Bojanowski et al. (2017) to get good initial values. The proposed model can handle both source-to-target translation and target-to-source translation like Johnson et al. (2017).

The training loss usually consists of two parts, a language model loss and a back-translation loss. Let $P_{\mathcal{X} \to \mathcal{Y}}$ denote the translation from space $\mathcal{X}$ to space $\mathcal{Y}$, and so for the other similar notations.

(1) Language model loss, implemented by a denoising autoencoder. Mathematically,

$$L_1 = \mathbb{E}_{x \sim \mathcal{X}}[- \log P_{\mathcal{X} \to \mathcal{X}}(x|C(x))] + \mathbb{E}_{y \sim \mathcal{Y}}[- \log P_{\mathcal{Y} \to \mathcal{Y}}(y|C(y))], \tag{12}$$

where $C(\cdot)$ is a noise model with randomly dropping several words, swapping words, etc.

(2) Back translation loss, implemented by back-translating the monolingual data and feeding into the reversed models. Mathematically,

$$L_2 = \mathbb{E}_{x \sim \mathcal{X}}[- \log P_{\mathcal{Y} \to \mathcal{X}}(x|\hat{y}(x))] + \mathbb{E}_{y \sim \mathcal{Y}}[- \log P_{\mathcal{X} \to \mathcal{Y}}(y|\hat{x}(y))], \tag{13}$$

in which $\hat{y}(x) = \arg\max_{u \in \mathcal{Y}} P_{\mathcal{X} \to \mathcal{Y}}(u|x)$ and $\hat{x}(y) = \arg\max_{v \in \mathcal{X}} P_{\mathcal{Y} \to \mathcal{X}}(v|y)$. Note that the four $P_{...}$'s are implemented in a single encoder-decoder based model, where each translation task has a different "task embedding", i.e., a learnable vector indicating the translating directions.

## D   IMAGE-TO-IMAGE TRANSLATION

### D.1   NETWORK ARCHITECTURE

We follow Zhu et al. (2017) to configure the model architectures for all tasks on image-to-image translation. For the generator, the network contains 2 stride-2 convolutions, 9 residual blocks, and 2 convolutional layers with stride $\frac{1}{2}$, (i.e., the transposed convolutional layers). Instance normalization is applied to the network. For the discriminator, the $70 \times 70$ PatchGANs (Isola et al., 2017), which aim to classify whether $70 \times 70$ overlapping image patches are real or fake. Such a patch-level discriminator architecture has fewer parameters than a full-image discriminator and can work on arbitrarily-sized images in a fully convolutional fashion.

### D.2   RESULTS OF PHOTO-TO-PAINTING TRANSLATION

We present results on photo→painting translation in Figure 5. As we can see, both standard Cycle-GAN and our algorithm successfully translates the inputs to the corresponding target domains. Our model outperforms the baseline in two aspects: 1) multi-agent dual learning can generate clearer images with less block artifact (the Monet column and Ukiyo-e column); and 2) the images generated by our algorithm are faithful to the original images in semantics. For example, the sun is missing from CycleGAN generated images in the top Ukiyo-e column and the bottom Cezanne column, while our algorithm is able to keep the sun.

### D.3   RESULTS OF CITYSCAPES PHOTO-TO-LABEL TRANSLATION

**FCN Evaluation** FCN-scores are used in Zhu et al. (2017) and Isola et al. (2017) to quantitatively evaluate Cityscape photo to semantic label translation. A pre-trained fully convolutional network (Long et al., 2015), briefly, FCN, is used to convert the photo to the corresponding semantic label, and then calculate the per-pixel accuracy, per-class accuracy and mean class Intersection-Over-Union(Class IOU) (Cordts et al., 2016) against the ground truth labels.

**Results on photo→label translation** The example cases and quantitative evaluation with FCN-scores of photo→labels on cityscapes images are shown in Figure 6 and Table 8[10] respectively. We

---

[10]The baseline performance is different from  Zhu et al. (2017) since it is unclear how the RGB based labels is converted to integer labels without the official evaluation code released. This is a common issue at `https://github.com/phillipi/pix2pix/issues/115`.

can observe improvements but the gap is not significant. Note that CycleGAN is a very strong baseline, and we leave further improving photo→label translation to future work.

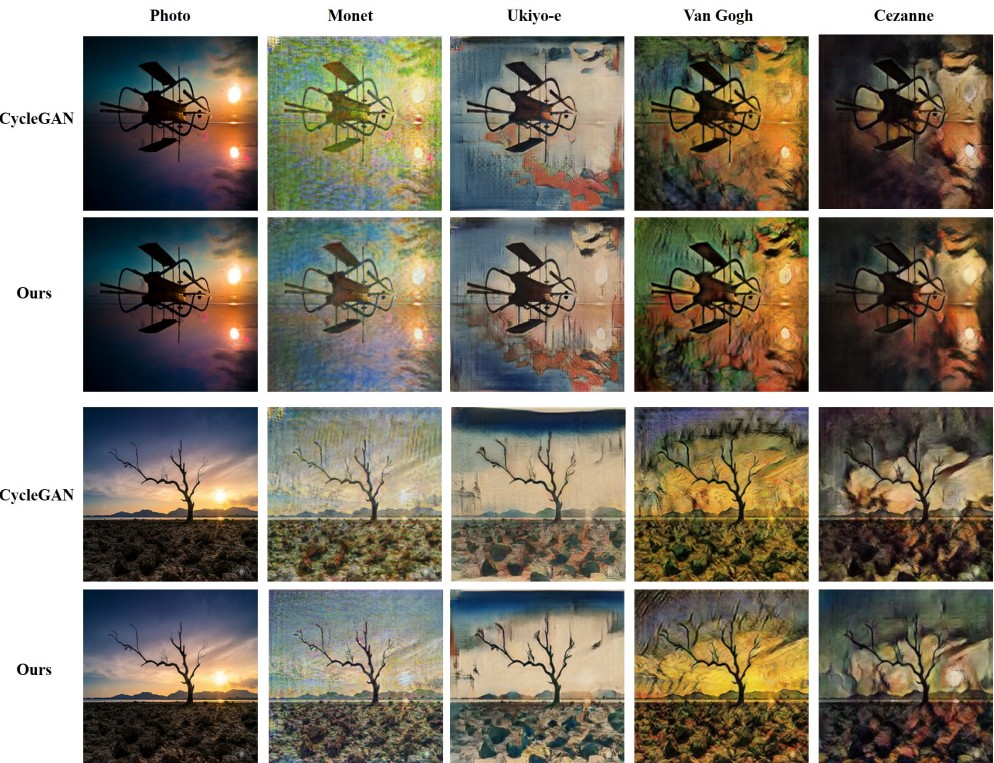

Figure 5: Two groups of experiments on photo→painting translations.

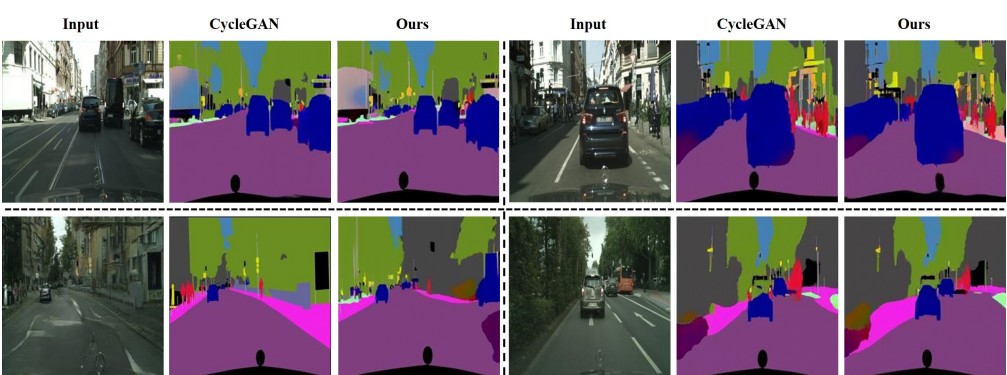

Figure 6: Cityscapes photo → labels translation

|          | Per-pixel Acc. | Per-class Acc. | Class IOU |
|----------|----------------|----------------|-----------|
| Standard | 0.56           | 0.19           | 0.15      |
| Dual-3   | 0.57           | 0.20           | 0.15      |

Table 8: Classification performances of Cityscapes photo → labels translation.

