# OpenReview forum: "Multi-Agent Dual Learning"
_ICLR.cc/2019/Conference_

### Official Review · AnonReviewer3 · 2018-11-02
**Extensive experiments and results, but not enough contribution**

**Rating:** 6
**Confidence:** 3

**Review:**

Summary:
The authors propose an extension of dual learning (DL). In DL, one leverages the duality of a dataset, by predicting both forward and backward, e.g. English to German, and German back to English. It’s been shown that training models using this duality is beneficial. This paper extends DL by introducing multiple models for the forward and backward, and using their output to regularise the training of the two main agents.

The authors show that this setup improves on the SotA, at only a training computation expense (inference/test time remains the same).

Review:
The paper shows extensive experimentation and improves the previous result in all cases. The proposed method is a straightforward extension and can be readily implemented and used.

I have difficulty understanding equation 8 and the paragraph below. It seems like the authors use an equal weighting for the additional agents, however they mention using Monte Carlo to “tackle the intractability resulting form the summation over the exponentially large space y”. According to the paper the size of y is the dataset, is it exponentially large? Do the authors describe stochastic gradient descent? Also what do the authors mean by offline sampling? Do they compute the targets for f_0 and g_0 beforehand using f_1…n and g_1…n?

The results mention computational cost a few times, I was wondering if the authors could comment on the increase in computational cost? e.g. how long does “pre-training” take versus training the dual? Can the training of the pre-trained agents be parallelised? Would it be possible to use dropout to more computationally efficient obtain the result of an ensemble?

In general I think the authors did an excellent job validating their method on various different datasets. I also think the above confusions can be cleared up with some editing. However the general contribution of the paper is not enough, the increase in performance is minimal and the increased computational cost/complexity substantial. I do think this is a promising direction and encourage the authors to explore further directions of multi-agent dual learning.

Textual Notes:
- Pg2, middle of paragraph 1: “which are pre-trained with parameters fixed along the whole process”. This is unclear, do you mean trained before optimising f_0 and g_0 and subsequently held constant?
- Pg2, middle last paragraph: “typical way of training ML models”. While the cross entropy loss is a popular loss, it is not “typical”.
- Pg 3, equation 4, what does “briefly” mean above the equal sign?
- Perhaps a title referring to ensemble dual learning would be more appropriate, given the possible confusion with multi agent reinforcement learning.


################
Revision:

I would like to thank the authors for the extensive revision, additional explanations/experiments, and pointing out extensive relevant literature on BLUE scores. The revision and comments are much appreciated. I have increased my score from 4 to 6.

---

> ### Author Response · Authors · 2018-11-21
> **Response to AnonReviewer3 [2/2]**
>
> ** Controlling Complexity **
> In this paper, we focus on demonstrating the effectiveness of our proposed method, while the issue of efficiency is not yet well explored. We agree with you that training efficiency is indeed also a very important issue. Setting a reasonable number of agents as we did in the paper is one way to control the complexity within a tolerable level while obtaining substantial gain.
>
> According to your comments, we further present a simple yet effective strategy to minimize the training complexity without too much loss in performance -- by generating different agents from a single run with warm restart. Specifically, we work with the following two settings:
>
> 1. Warm restart by learning rate schedule.
> (a) Setting: We employ the warm restart strategy in [1], where the warm restart is emulated by increasing the learning rate. Specifically, learning rate starts from an initial value L, then decays with a cosine annealing. Once a cyclic iteration is reached, the learning rate is increased back to the initial value and then followed with cosine decay. At the end of each cycle where the learning rate is of the minimal value, the model is approximately a local optimal. Thus, we can use multiple different such local optima as our agents.
>  (b) Pre-training Cost: Training one agent on IWSLT takes 3 days on 1 GPU (i.e. 3 GPU days). Thus, for Dual-5 model which involves 4 additional pairs of agents, the total pre-training cost in our original way through independent runs is 4 (pairs) * 2 (directions: De->En and En->De) * 3, in total 24 GPU days. With the new learning rate schedule, we can obtain the 4 pairs of agents with a single run which takes 2 (directions: De->En and En->De) * 3, in total 6 GPU days. Such a method is three times more efficient than the original way.
> (c) Performance: With this strategy, we are able to achieve 35.07 and 29.40 BLEU with Dual-5 on IWSLT De->En and En->De respectively.  Although not as good as our original method with higher complexity (e.g., 35.44 BLEU in De->En and 29.52 BLEU in En->De), such light-weighted version of our method is still able to outperform the baselines with large margin for over 1 BLEU score with minimal increase in training cost.
>
> 2. Warm restart with different random seeds and training subsets.
> (a) Setting: We first train a model to a stage that the model is not converged but has relatively good performance. We then use this model as warm start, and train different agents with different iteration over the dataset and different subsets. This strategy intuitively works better with larger dataset. We present results in WMT En<->Fr translation.
> (b) Pre-training Cost: Training one agent on WMT En<->Fr dataset takes 7 days on 8 GPUs, in total 56 GPU days. For Dual-3 with 2 additional pairs of agents, the total pre-training cost is 2 * 2 * 56 = 224 GPU days. With the above strategy, we managed to decrease the cost into 2 * 56 + 2 * 8 = 128 GPU days.
> (c) Performance: We are able to achieve 43.87 BLEU and 40.14 with Dual-3 on WMT En-Fr and Fr-En respectively, which improves 1.37 and 1.74 points over the baselines (42.5 for En->Fr and 38.4 for Fr->En).
>
> With the above two strategies, we demonstrate that our framework is also capable of improving performance with large margin while introducing minimal computational cost. We will definitely further study the best strategy to minimize the training complexity while maintaining the improvements in our future work.
>
> ** Textual Notes **
> Thanks for pointing it out. We edit the writing in our updated paper.
> Although with the same term, the "multi-agent" in this paper has no relationship with multi-agent reinforcement learning. To avoid further confusion in the discussion period, currently we decide not to change the paper title during rebuttal.
>
> We hope the above explanations could address your concerns. Please kindly check our updated paper with clarification and new experimental results.
> Thanks for your time and valuable feedbacks.
>
> [1] Loshchilov, Ilya, and Frank Hutter. "Sgdr: Stochastic gradient descent with warm restarts." In Proc. of ICLR, 2017.
> [2] Peter Shaw, Jakob Uszkoreit, and Ashish Vaswani. Self-attention with relative position representations. In Proc. of NAACL, 2018.
> [3] Anonymous. Universal transformers. In Submitted to International Conference on Learning Representations, 2019. URL https://openreview.net/forum?id=HyzdRiR9Y7. Under review as a conference paper at ICLR 2019

---

> ### Author Response · Authors · 2018-11-21
> **Response to AnonReviewer3 [1/2]**
>
> Summary: our response includes (1) Clarification on Equation 8 and its descriptions; (2) Explanations on computational cost; (3) Clarification on contribution and (4) Discussion on controlling complexity.
>
> ** Equation 8 and its descriptions **
> We apologize for the confusions with equation 8. We have reorganized Section 3.1 in the update paper. To answer your questions:
> 1. Space Y: Space \mathcal{Y} refers to the collection of all possible sentences of the Y domain language, instead of just the dataset (denoted by D_y, where we have D_y \in \mathcal{Y}). That's why it could be exponentially large.
> 2. Offline sampling: We do offline sampling by sampling all the x_hat and y_hat with f_i and g_i respectively in advance (for i>=1). We reorganized Section 3.1 and Algorithm 1 to more clearly explain how to estimate the gradients and do the offline sampling.
>
> ** Explanations on Computational Cost **
> The computational cost refers to GPU time for training. Although pre-training can be parallelized, the total GPU time will not be reduced. For example, on WMT14 En<->De task, it takes 40 GPU days (5 days on 8 GPU) to train one model (agent). Pre-training more agents takes more GPU time with either more GPUs to train in parallel or longer training time. This is what we mean by "increased computational cost" with more agents.
> However, as is shown from our experiments, we can obtain significant improvements over the strong baseline models with multiple but not too much agents (e.g. with n=3, which brings tolerable increase in computational cost yet substantial gain). Note that we do not increase the computational cost during inference.
>
> ** Contribution & Improvement **
> We propose a new multi-agent dual learning framework that leverages more than one primal models and dual models in the learning system. Our framework has demonstrated its effectiveness on multiple machine translation and image translation tasks:
> 1. We work on six NMT tasks to evaluate our algorithm (see Section 3).  Our improvement over the strong baselines with the state-of-the-art transformer model is not minimal. As can be seen from the recent literature in NMT [2][3], transformer is a powerful and robust model, and improving BLEU by 1 point over such strong baseline is generally considered as a non-trivial progress. Our method yields consistent and substantial improvement across all the benchmark datasets.
> 2. Our method is capable of further improving the state-of-the-art model. We work on WMT18 English-to-German translation tasks, and achieve a 49.61 BLEU score, which outperforms the champion system by 1.31 point and sets a new record on this task (see Table 4 in Section 3.4 of our updated paper).
> 3. Our method also works for unsupervised image generation. We achieve consistent improvements over CycleGAN quantitatively and qualitatively (See Section 4).

---

> ### Author Response · Authors · 2018-11-28
> **Further response to AnonReviewer3**
>
> Dear AnonReviewer3:
>
> Thanks for your response to our rebuttal. However, it is unclear to us why you believe that the general contribution of the paper remains too small for ICLR because of the subjectivity of your criticism. What does "too small" mean exactly?
>
> Our best interpretation of your concern is "the increase in performance is minimal and the increased computational cost/complexity substantial". While this is a legitimate concern, we do not believe the concern is sufficiently substantial to justify a rating of the paper below the acceptance threshold for the following reasons:
>
> 1. "The increase in performance is minimal ":
> While the performance improvement may appear to be small, it is known that the improvement of BLEU score is difficult, and the magnitude of improvement from our methods is better than or at least comparable to the reported improvement on this task by recent papers published in major venues such as NeurIPS. For example, on the WMT2014 En->De translation task, the performance of the transformer baseline is 28.4 BLEU score [1] (our baseline matches this performance). The improvement over this baseline is 0.61 in [2], 0.8 in [3] (1.3 BLEU improvement over the re-implemented 27.9 baseline in [3]) and 0.9 in [4], while ours is 1.65 BLEU score. We perform paired bootstrap sampling [5] for significance test using the script in Moses [6]. Our improvement over the baselines are statistically significant with p < 0.01 across all machine translation tasks.
> Moreover, as we pointed out in the previous response, our method has achieved the best performance so far on IWSLT 2014 De->En and WMT 2018 En->De. Our main point here is that our experimental results have provided solid evidence that the proposed new method has clearly advanced the state of the art on multiple tasks.
>
> 2. "The increased computational cost/complexity substantial":
> As we already explained in our previous response, the computational complexity can be further reduced (there are potentially other ways to further improve efficiency), so this is not an *inherent* deficiency of the proposed new approach, but rather interesting new research questions that can be further investigated in the future.  Thus in this sense, our work has also opened up some new interesting research directions.
>
> We welcome further discussion and are willing to answer any further questions.
>
>
> [1] Vaswani, Ashish, et al. "Attention is all you need." Advances in Neural Information Processing Systems. 2017.
> [2] He, Tianyu, et al. "Layer-Wise Coordination between Encoder and Decoder for Neural Machine Translation". Advances in Neural Information Processing Systems. 2018.
> [3] Shaw, Peter, Jakob Uszkoreit, and Ashish Vaswani. "Self-Attention with Relative Position Representations." In Proc. of NAACL, 2018.
> [4] Anonymous. Universal transformers. In Submitted to International Conference on Learning Representations, 2019. URL https://openreview.net/forum?id=HyzdRiR9Y7. Under review as a conference paper at ICLR 2019
> [5] Koehn, Philipp. "Statistical significance tests for machine translation evaluation." Proceedings of the 2004 conference on empirical methods in natural language processing. 2004.
> [6] https://github.com/moses-smt/mosesdecoder/blob/master/scripts/analysis/bootstrap-hypothesis-difference-significance.pl

---

> > ### Comment · AnonReviewer3 · 2018-11-28
> > **Calibration of score**
> >
> > Dear Authors,
> >
> > Thank you for pointing out the extensive relevant literature. I had indeed underestimated the improvement in BLUE score and will update my score to a 6.

---

### Official Review · AnonReviewer1 · 2018-11-03
**Applying ensembles to machine translation appears to result in good performance on language and image translation**

**Rating:** 6
**Confidence:** 2

**Review:**

The author's present a dual learning framework that, instead of using a single mapping for each mapping task between two respective domains, the authors learn multiple diverse mappings. These diverse mappings are learned before the two main mappings are trained and are kept constant during the training of the two main mappings. Though I am not familiar with BLEU scores and though I didn't grasp some of the details in 3.1, the algorithm yielded consistent improvement over the given baselines. The author's included many different experiments to show this.

The idea that multiple mappings will produce better results than a single mapping is reasonable given previous results on ensemble methods.

For the language translation results, were there any other state-of-the-art methods that the authors could compare against? It seems they are only comparing against their own implementations.

Objectively saying that the author's method is better than CycleGAN is difficult. How does their ensemble method compare to just their single-agent dual method? Is there a noticeable difference there?

Minor Comments:

Dual-1 and Dual-5 are introduced without explanation.

Perhaps I missed it, but I believe Dan Ciresan's paper "Multi-Column Deep Neural Networks for Image Classification" should be cited.

### After reading author feedback
Thank you for the feedback. After reading the updated paper I still believe that 6 is the right score for this paper. The method produces better results using ensemble learning. While the results seem impressive, the method to obtain them is not very novel; nonetheless, I would not have a problem with it being accepted, but I don't think it would be a loss if it were not accepted.

---

> ### Author Response · Authors · 2018-11-21
> **Response to AnonReviewer1**
>
> Thank you for your review and valuable comments!
>
> Summary: our response includes: (1) Clarification on language translation baselines; (2) Discussion on image translation evaluation; (3) Reference and clarification.
>
> ** Language Translation Baselines **
> 1. For the baseline models reported:
> 1.1) We use the transformer model with "transformer_big" setting [1], which is a strong baseline that outperforms almost all previously popular NMT models based on CNN [2] and LSTM [3]. Transformer is the state-of-the-art NMT architecture. Our numbers of the baseline transformer model match the results reported in [1].
> 1.2) In addition to the standard baseline models, we also compare our method against all the relevant algorithms including knowledge distillation (KD) and back translation (BT).
> 1.3) As can be seen in many well-known and recent NMT works ([4], [5]), it is a common practice to use transformer as the robust baseline model. Furthermore, it is also shown from these works that it is hard to improve over the transformer baseline, and 0.5-1 BLEU score improvement is already considered substantial.
>
> 2. We further add newly obtained results on the WMT18 challenge. We compare our method with both the champion translation system MS-Marian (WMT18 En->De challenge champion). Our method achieves the state-of-the-art result on this task.
> ---------------------------------------------------------------------------
>  WMT En->De                     2016          2017        2018
> ---------------------------------------------------------------------------
> MS-Marian (ensemble)     39.6           31.9          48.3
> Ours (single)                       40.68         33.47       48.89
> Ours (ensemble)                41.23         34.01       49.61
> ---------------------------------------------------------------------------
> Please refer to Section 3.4 "Study on generality of the algorithm" for more details and Table 4 for full results in our updated paper.
>
> ** Image Translation Evaluation **
> For image-to-image translation tasks, we further add two quantitative measures: (1) We use the Fréchet Inception Distance (FID) [6], which measures the distance between generated images and real images to evaluate the painting to photos translation. (2) We use "FCN-score" evaluation on the cityscape dataset following [7]. The results are reported in Table 6 and Table 7 respectively. Multi-agent dual learning framework can achieve better quantitative results than the baselines.
>
> We are not sure what you meant by “How does their ensemble method compare to just their single-agent dual method?”. The standard CycleGAN model (baseline) already leverages both primal and dual mappings, which is equivalent to our “Dual-1” model in NMT experiments, i.e., the dual method with only one pair of agents f_0 and g_0. Our model involves two additional pairs of agents (f_1 and g_1, f_2 and g_2) during training. Unlike ensemble learning, only one agent (f_0 for forward direction, or g_0 for backward direction) is used during inference.
>
> ** Reference **
> Thanks for pointing a reference paper "Multi-Column Deep Neural Networks for Image Classification" (briefly, MCDNN) and we have added reference to it (Section 4).
> Although MCDNN also uses multiple agents (i.e., several columns of deep neural networks), it differs from our model in two aspects: (1) Our work leverages the duality of a pair of dual tasks while this paper does not; (2) In an MCDNN framework, during the training phase, all the columns are updated by winner-take-all rule; and during inference, all columns work like an ensemble model through weighted average. In comparison, we only update one primal and one dual agent during training, and use one agent for inference.
>
> ** Clarity **
> Thanks for pointing out that our original introduction to the names of baselines and models is not very clear. Please kindly refer to first paragraph in Section 3.3.
>
> You may check our updated paper with clarification and new experimental results.
> Thanks for your time and feedbacks.
>
> [1] Vaswani, Ashish, et al. "Attention is all you need." In NIPS. 2017.
> [2] Jonas Gehring, Michael Auli, David Grangier, Denis Yarats, and Yann N Dauphin. Convolutional Sequence to Sequence Learning. In Proc. of ICML, 2017.
> [3] Wu, Yonghui, et al. "Google's neural machine translation system: Bridging the gap between human and machine translation." arXiv preprint arXiv:1609.08144 (2016).
> [4] Chen, Mia Xu, et al. "The Best of Both Worlds: Combining Recent Advances in Neural Machine Translation." In Proc. of the ACL, 2018.
> [5] Peter Shaw, Jakob Uszkoreit, and Ashish Vaswani. Self-attention with relative position representations. In Proc. of NAACL, 2018.
> [6] Heusel, Martin, et al. "Gans trained by a two time-scale update rule converge to a local nash equilibrium." In NIPS, 2017.
> [7] Isola, Phillip, et al. "Image-to-image translation with conditional adversarial networks." In CVPR, 2017

---

> ### Author Response · Authors · 2018-12-11
> **Further response to AnonReviewer1**
>
> Dear AnonReviewer1,
>
> Before the final decision concludes, do you have further questions regarding our rebuttal and updated paper? Our paper revision includes reorganization of the introduction to our framework (Section 3.1), the additional experiments on WMT18 English->German translation challenge (Section 3.4), the additional study on diversity of agents (Appendix A), and quantitative evaluation on image-to-image translations (Section 4.3 and 4.4) following your suggestions.
>
> In particular, we would like to highlight that:
> (1) The calibration of BLEU score: We would like to point out that our improvement over the previous state-of-the-art baselines is substantial. For example, on the WMT2014 En->De translation task, the performance of the transformer baseline is 28.4 BLEU score [1] (our baseline matches this performance). The improvement over this baseline is 0.61 in [2], 0.8 in [3] (1.3 BLEU improvement over the re-implemented 27.9 baseline in [3]) and 0.9 in [4], while ours is 1.65 BLEU score.
> (2) The baselines: As we explained in the previous response, we are using the state-of-the-art transformer as our backbone model, and comparing against all the relevant algorithms including KD, BT and the traditional 2-agent dual learning (Dual-1). Moreover, we also show on WMT18 En->De challenge that our method can further improve the state-of-the-art model trained with extensive resources (Section 3.4 of our updated paper).
>
> We hope our rebuttal and paper revision could address your concerns. We welcome further discussion and are willing to answer any further questions.
>
> [1] Vaswani, Ashish, et al. "Attention is all you need." Advances in Neural Information Processing Systems. 2017.
> [2] He, Tianyu, et al. "Layer-Wise Coordination between Encoder and Decoder for Neural Machine Translation". Advances in Neural Information Processing Systems. 2018.
> [3] Shaw, Peter, Jakob Uszkoreit, and Ashish Vaswani. "Self-Attention with Relative Position Representations." In Proc. of NAACL, 2018.
> [4] Anonymous. Universal transformers. In Submitted to International Conference on Learning Representations, 2019. URL https://openreview.net/forum?id=HyzdRiR9Y7. Under review as a conference paper at ICLR 2019

---

### Official Review · AnonReviewer2 · 2018-11-03
**Straightforward Idea, pretty good results, some things should be clarified (potential issue with the maths).**

**Rating:** 6
**Confidence:** 4

**Review:**

Summary

The paper proposes to modify the "Dual Learning" approach to supervised (and unsupervised) translation problems by making use of additional pretrained mappings for both directions (i.e. primal and dual). These pre-trained mappings ("agents") generate targets from the primal to the dual domain, which need to be mapped back to the original input. It is shown that having >=1 additional agents improves training of the BLEU score in standard MT and unsupervised MT tasks. The method is also applied to unsupervised image-to-image "translation" tasks.

Positives and Negatives
+1 Simple and straightforward method with pretty good results on language translation.
+2 Does not require additional computation during inference, unlike ensembling.
-1 The mathematics in section 3.1 is unclear and potentially flawed (more below).
-2 Diversity of additional "agents" not analyzed (more below).
-3 For image-to-image translation experiments, no quantitative analysis whatsoever is offered so the reader can't really conclude anything about the effect of the proposed method in this domain.
-4 Talking about "agents" and "Multi-Agent" is a somewhat confusing given the slightly different use of the same term in the reinforcement literature. Why not just "mapping" or "network"?

-1: Potential Issues with the Maths.

The maths is not clear, in particular the gradient derivation in equation (8). Let's just consider the distortion objective on x (of course it also applies to y without loss of generality). At the very least we need another "partial" sign in front of the "\delta" function in the numerator. But again, it's not super clear how the paper estimates this derivative.  Intuitively the objective wants f_0 to generate samples which, when mapped back to the X domain, have high log-probability under G, but its samples cannot be differentiated in the case of discrete data. So is the REINFORCE estimator used or something? Not that the importance sampling matter is orthogonal. In the case of continuous data x, is the reparameterization trick used? This should at the very least be explained more clearly.

Note that the importance sampling does not affect this issue.

-2: Diversity of Agents.

As with ensembles, clearly it only helps to have multiple agents (N>2) if the additional agents are distinct from f_1 (again without loss of generality this applies to g as well). The paper proposes to use different random seeds and iterate over the dataset in a different order for distinct pretrained f_i. The paper should quantify that this leads to diverse "agents". I suppose the proof is in the pudding; as we have argued, multiple agents can only improve performance if they are distinct, and Figure 1 shows some improvement as the number of agents are increase (no error bars though). The biggest jump seems to come from N=1 -> N=2 (although N=4 -> N=5 does see a jump as well). Presumably if you get a more diverse pool of agents, that should improve things. Have you considered training different agents on different subsets of the data, or trying different learning algorithms/architectures to learn them? More experiments on the diversity would help make the paper more convincing.

---

> ### Author Response · Authors · 2018-11-21
> **Response to AnonReviewer2**
>
> Thank you for your comments and suggestions!
>
> Summary: our response includes (1) Clarification on math equations; (2) Analysis on diversity of additional agents; (3) Quantitative analysis for image translation.
>
> ** Clarification on Mathematics in Section 3.1 **
> We apologize for the confusions in Section 3.1. We have reorganized this section, as shown in our updated paper. For your questions:
> 1. About equation 8, indeed there is a typo and should be a "partial" sign in front of the "\delta" function in the numerator. Thanks for pointing this out.
> 2. The details of derivative estimation can be found in Section 3.1 (especially equation 9 and 10 in our updated version.
>
> ** Study on diversity of agents **
> 1. You are right. We obtained distinct "agents" f_i and g_i through multiple independent runs with different random seeds and different input orders of the training samples. As far as we know, there's no common quantitative metric to measure the diversity among models in NMT. But we agree with you that intuitively, more diversity among agents leads to greater improvements.
>
> 2. Following your suggestions, we add a study on the diversity of agents (presented in Appendix A of the updated paper). We design three group of agents with different levels of diversity: (E1) Agents with the same network structure trained by independent runs, i.e., what we use in Section 3.3; (E2) Agents with different architectures and independent runs; (E3) Homogeneous agents of different iteration, i.e., the checkpoints obtained at different (but close) iterations from the same run. We evaluate the above three settings on IWSLT2014 De<->En dataset. The diversity of the above three settings would intuitively be (E2)>(E1)>(E3). We present full results in Figure 4 (Appendix A), where the BLEU scores with Dual-5 model are:
>
> --------------------------------------------------------
>                             E1             E2             E3
> --------------------------------------------------------
> En -> De          35.44       35.56       34.97
> De -> En          29.52       29.58       29.28
> --------------------------------------------------------
>
> From the above results, we can see that diversity among agents indeed plays an important role in our method. There are, of course, many other ways to introduce more diversity, including using different optimization strategies, or training with different subsets as you suggested. All of these can potentially bring further improvements to our framework, yet are not the focus of this work. From the current studies, we show that our algorithm is able to achieve substantial improvement with a reasonable level of diversity. We leave more comprehensive studies on diversity to future work.
>
> Please kindly refer to Appendix A for more detailed results.
>
> ** Quantitative analysis for image translation **
> Thanks for your suggestions. We add two quantitative measures on image translation tasks: (1) We use the Fréchet Inception Distance (FID score) [1], which measures the distance between generated images and real images to evaluate the painting to photos translation. (2) We use "FCN-score" evaluation on the cityscape dataset following [2]. The results are reported in Table 6 and Table 7 respectively. Multi-agent dual learning framework can achieve better quantitative results than the baselines.
>
> ** Term usage of "multi-agents" **
> Although with the same term, the "multi-agent" or "agent" in this paper has no relationship with multi-agent reinforcement learning. You are right in that the term "agent" in our context refers to "mapping" or "network". To avoid further confusion in the discussion period, currently we decide not to change the term usage throughout the paper during rebuttal; instead, we will change the term after the acceptance/rejection decision.
>
> You can check our updated paper with clarification and new experimental results.
> Thanks for your time and valuable feedbacks.
>
> [1] Heusel, Martin, et al. "Gans trained by a two time-scale update rule converge to a local nash equilibrium." Advances in Neural Information Processing Systems. 2017.
> [2] Isola, Phillip, et al. "Image-to-image translation with conditional adversarial networks." In CVPR, 2017

---

### Public Comment · (anonymous) · 2018-10-31
**some details**

What's the batch size of your baseline system for IWSLT De-En? And which evaluation script do you use to measure the BLEU score?

I run the T2T with transform_base parameters(batch size = 320), and achieve a BLEU score of 34.38, which is higher than your baseline (33.42). I use the multi_bleu.pl and tokenize the English and German using Moses toolkit.

---

> ### Author Response · Authors · 2018-10-31
> **Reply to some details**
>
> Thanks for your comments.
>
> For IWSLT De-En, we use the 'transformer_small' setting (in paper section 3.2), in which the batch size is set to be 4096 tokens. We use multi-bleu.pl to evaluate the tokenized BLEU.
>
> Thanks for providing a stronger baseline and we are working on it. To confirm, by 'batch size=320', are you referring to 320 tokens or sentences?

---

> > ### Public Comment · (anonymous) · 2018-11-02
> > **batch size**
> >
> > Thanks for your reply, I used 320 tokens to obtain a better result compared to the default settings.

---

> > > ### Author Response · Authors · 2018-11-02
> > > **IWSLT De->En experiment details**
> > >
> > > Thanks for the information. Here are our settings and some initial observations:
> > >
> > > ** Settings **
> > >   -  Hyperparameters: We set 'hparams_set=transformer_base', and experiment with the batch size of 4096 (default), 6400 (to approximate 320 sentences) and 320 tokens, and dropout rate of 0.1 (default) and 0.4 (since severe overfitting observed). The rest hyperparameters use the default value in 'transformer_base'.
> > >   -  Optimization: We use the Adam optimizer with the same setting described in the paper (section 3.2 Optimization and Evaluation).
> > >   -  Evaluation: We use beam search with a beam size of 6 (paper section 3.2) in inference and use multi-bleu.pl to evaluate the tokenized BLEU.
> > >
> > > We run the baseline and our algorithm with 5 agents (Dual-5) with the above settings. For our multi-agent model, we still use the same agents as the paper (transformer_small with 4 blocks) for sampling.
> > > The models are implemented with tensor2tensor v1.2.9 and trained on one M40 GPU.
> > >
> > > ** Results **
> > > Below are the initial results. We are still working on the experiments.
> > >
> > > 	Table 1. With dropout rate of 0.1 (default)
> > > 	-------------------------------------------------------------------------------
> > > 	Batch Size                   4096                6400               320
> > > 	-------------------------------------------------------------------------------
> > > 	Baseline                      32.24               32.22             2.17
> > > 	Ours (Dual-5)             34.59               34.58             3.65
> > > 	-------------------------------------------------------------------------------
> > >
> > > 	Table 2. With dropout rate of 0.4
> > > 	-------------------------------------------------------------------------------
> > > 	Batch Size                   4096                6400               320
> > > 	-------------------------------------------------------------------------------
> > > 	Baseline                      34.40               34.43             2.37
> > > 	Ours (Dual-5)             35.12               35.45             3.91
> > > 	-------------------------------------------------------------------------------
> > >
> > > We have the following observations:
> > > 	1) The default 'transformer_base' setting appears to suffer from severe overfitting (Table 1). We tune the dropout ratio and present results with dropout=0.4 in Table 2, where we indeed obtain better baseline results than our baselines with 'small' setting reported in the paper. The stronger baseline achieves a 34.43 BLEU score (with batch size 6400).
> > > 	2) We notice that with a batch size of 320 tokens (as the setting you suggested), the model is not well optimized with either dropout ratio. We are curious whether you are also using other different hyperparameters or optimization settings. We would be happy to re-evaluate our approach under the stronger baseline setting.
> > > 	3) From the results we have so far, our algorithm can still outperform the stronger baseline with a large margin, achieving 35.45 BLEU score (with batch size 6400).
> > >
> > > We will keep working on experiments of IWSLT De-En under the 'base' settings and update our findings.

---

### Author Response · Authors · 2018-11-26
**Paper revision and summary of contributions**


Dear Reviewers:

Thanks for the valuable comments and discussion.

Our paper revision seeks to clarify the introduction to our framework and strengthen the experiment results, which includes: (1) reorganization and clarification in Section 3.1; (2) the additional study on diversity of agents (Appendix A); (3) the additional experiment results on WMT18 English->German translation challenge (Section 3.4); and (4) quantitative evaluation on image-to-image translations (Section 4.3 - 4.4).

In particular, we would like to highlight our contribution in this work. We are the first to incorporate multiple agents into the dual learning framework, extending traditional dual learning to a much more general concept.  The multi-agent dual learning framework, which is generally applicable to many different tasks, has significantly pushed the frontier towards dual learning. In particular, we show how the proposed general framework can be adapted to the machine translation and image translation tasks. The method is non-trivial yet very easy to apply and has been proved to be very powerful across many different translation tasks with our extensive empirical studies:

1) Our proposed framework has achieved broad success: we have evaluated our method on five image-to-image translation tasks, and six machine translation tasks across different language pairs, different dataset scale (small dataset like IWSLT and large dataset like WMT) and different machine learning setting (supervised and unsupervised). Our method demonstrates consistent and substantial improvements over the standard baseline and traditional (two-agent) dual learning method.

2) The multi-agent dual learning framework also pushes forward the state-of-the-art performances. On IWSLT 2014 German->English translation, we set a new record of a 35.44 BLEU score. On the recent WMT 2018 English ->German translation, we achieve the state-of-the-art performance of a 49.61 BLEU score, outperforming the challenge champion by over 1.3 BLEU score.

We believe we have made decent contributions in this paper based on all these above. We welcome further discussion and are willing to answer any further questions.

Thanks,
The Authors

---

### Meta-Review · Area_Chair1 · 2018-12-14
**Accept**

**Confidence:** 4
**Recommendation:** Accept (Poster)

**Metareview:**

A paper that studies two tasks: machine translation and image translation. The authors propose a new multi-agent dual learning technique that takes advantage of the symmetry of the problem. The empirical gains over a competitive baseline are quite solid. The reviewers consistently liked the paper but have in some cases fairly low confidence in their assessment.